# Altered Mitochondrial Quality Control in Rats with Metabolic Dysfunction-Associated Fatty Liver Disease (MAFLD) Induced by High-Fat Feeding

**DOI:** 10.3390/genes13020315

**Published:** 2022-02-08

**Authors:** Federica Cioffi, Antonia Giacco, Giuseppe Petito, Rita de Matteis, Rosalba Senese, Assunta Lombardi, Pieter de Lange, Maria Moreno, Fernando Goglia, Antonia Lanni, Elena Silvestri

**Affiliations:** 1Department of Sciences and Technologies, University of Sannio, 82100 Benevento, Italy; federica.cioffi@unisannio.it (F.C.); agiacco@unisannio.it (A.G.); moreno@unisannio.it (M.M.); goglia@unisannio.it (F.G.); 2Department of Environmental, Biological and Pharmaceutical Sciences and Technologies, University of Campania “L. Vanvitelli”, 81100 Caserta, Italy; giuseppe.petito@unicampania.it (G.P.); rosalba.senese@unicampania.it (R.S.); pieter.delange@unicampania.it (P.d.L.); antonia.lanni@unicampania.it (A.L.); 3Department of Biomolecular Sciences, Urbino University, 61029 Urbino, Italy; rita.dematteis@uniurb.it; 4Department of Biology, University of Naples Federico II, 80126 Naples, Italy; aslombar@unina.it

**Keywords:** MAFLD, oxidative stress, mitochondrial DNA (mtDNA), mitochondrial quality control, mitophagy, mtBER, thermoneutrality, fatty liver disease

## Abstract

Metabolic dysfunction-associated fatty liver disease (MAFLD) is defined as the presence of hepatic steatosis in addition to one of three metabolic conditions: overweight/obesity, type 2 diabetes mellitus, or metabolic dysregulation. Chronic exposure to excess dietary fatty acids may cause hepatic steatosis and metabolic disturbances. The alteration of the quality of mitochondria is one of the factors that could contribute to the metabolic dysregulation of MAFDL. This study was designed to determine, in a rodent model of MAFLD, the effects of a long-term high-fat diet (HFD) on some hepatic processes that characterize mitochondrial quality control, such as biogenesis, dynamics, and mitophagy. To mimic the human manifestation of MAFLD, the rats were exposed to both an HFD and a housing temperature within the rat thermoneutral zone (28–30 °C). After 14 weeks of the HFD, the rats showed significant fat deposition and liver steatosis. Concomitantly, some important factors related to the hepatic mitochondrial quality were markedly affected, such as increased mitochondrial reactive oxygen species (ROS) production and mitochondrial DNA (mtDNA) damage; reduced mitochondrial biogenesis, mtDNA copy numbers, mtDNA repair, and mitochondrial fusion. HFD-fed rats also showed an impaired mitophagy. Overall, the obtained data shed new light on the network of different processes contributing to the failure of mitochondrial quality control as a central event for mitochondrial dysregulation in MAFLD.

## 1. Introduction

MAFLD is defined as the presence of hepatic steatosis in addition to one of three metabolic criteria: overweight/obesity, type 2 diabetes mellitus (T2DM), or metabolic dysregulation. MAFLD affects about a quarter of the world's adult population, thus representing a major health and economic burden to all societies. Currently, it still does not have an approved drug therapy [1,2,3,4]. Research aimed at investigating the cellular/molecular mechanisms involved in the genesis of this disease are therefore of considerable interest. 

Previous studies have shown that animals fed for several weeks on an HFD, with a fat content of more than 40%, can develop obesity, hyperinsulinemia, hyperglycemia, hypertension, and liver damage, i.e., a similar phenotype to that observed in humans with MAFLD/NAFLD [5,6,7,8,9]. Several studies have suggested hepatic mitochondrial dysfunction as an etiological mechanism [10,11,12,13,14,15]. HFD feeding increases the amount of substrate available for oxidation and, at the same time, the electron flux through the mitochondrial respiratory chain, enhancing free radical production [16]. In addition, HFD may produce an impairment of the antioxidant defenses [7]. When mitochondrial ROS are chronically produced at high levels, they can modify cellular functional macromolecules, such as lipids, proteins, and DNA [17]. Because of its close proximity to the major cell site of ROS generation and its paucity of protective histones, mitochondrial DNA (mtDNA) is highly susceptible to oxidative stress-induced damage [18,19]. Recent research has shown that the liver mtDNA of NAFLD patients has a higher mutational rate in comparison to that of healthy people [19]. Furthermore, in the past decade, numerous studies have highlighted mitochondrial dysfunction as one of the major contributors of several metabolic disorders, such as diabetes and obesity, diseases in which the accumulation of dysfunctional mitochondria leads to oxidative stress and impaired cell functions [20]. In this context, three processes attracted particular attention, namely mitochondrial dynamics (fusion and fission), biogenesis, and mitophagy, all identified as key determinants of mitochondrial quality control [21,22,23]. Specifically, increasing lines of evidence suggest that mitophagy plays an important role in degrading damaged mitochondria in the liver, both at baseline and/or in response to oxidative stress. 

One of the problems in correctly translating the results of metabolic disease studies in rodents to biomedical data for humans arises from the housing temperature used in the animal modeling, which is crucial due to the existing differences between animal and human thermal physiology [24]. Rats are often used as models of human physiology/pathology and are routinely housed at standard temperatures (19–22 °C) that are actually far below their thermoneutral zone (28–30 °C); Subsequently, they are cold-exposed and activate all the metabolic processes useful to counteract heat loss. Subthermoneutral stress is not common in daily life human habitations. For this reason, thermoneutral housing should be used for the modeling of human metabolic studies in rodents, and it is important for the correct comparison of metabolic data to take into consideration any physiological modification due to housing temperature [24,25,26]. 

In the current study, we examined the relevance of the putative alterations of liver mitochondria in a MAFLD rat model, in which a “human-like” MAFLD was induced by HFD-feeding for 14 weeks in association with thermoneutral housing. In particular, changes in mitochondrial ROS production, DNA damage and repair, and mitochondrial quality control mechanisms (i.e., biogenesis, dynamics, and mitophagy) were investigated.

## 2. Materials and Methods

### 2.1. Animals

Male Wistar rats (Rattus Norvegicus, 250–300 g, aged eight weeks) were maintained one per cage in a temperature-controlled room at 28 °C (thermoneutrality for rats) under a 12 h dark/light cycle and housed with unlimited access to water. All animals received humane care according to the criteria outlined in the *Guide for the Care and Use of Laboratory Animals* prepared by the National Academy of Sciences and published by the National Institutes of Health. All animal protocols were approved by the Committee on the Ethics of Animal Experiments of the University of Campania “L.Vanvitelli” (Italy) and the Italian Ministry of Health (permit number: 704/2016-PR of the 15 July 2016; project number: 83700.1 of the 3 May 2015). Every effort was made to minimize animal pain and suffering. At the start of the study (day 0), and after seven days of acclimatization to thermoneutrality, the rats were divided into two groups, as follows:The first group, control group (N), received a standard diet ad libitum (total metabolizable percentage of energy: 60.4 carbohydrates, 29 proteins, and 10.6 fat J^−1^; 15.88 kJ gross energy g^−1^) (Muscedola, Milan, Italy) for fourteen weeks;The second group, MAFLD group (HFD) received a high-fat diet (HFD) ad libitum (280 g diet supplemented with 395 g of lyophilized lamb meat (Liomellin, Milan, Italy), 120 g cellulose (Sigma-Aldrich, St. Louis, MO, USA), 20 g mineral mix (ICN Biomedical, Solon, OH, USA), 7 g vitamin mix (ICN), and 200 g low-salt butter (Lurpak, Denmark); total metabolizable percentage of energy: 21 carbohydrates, 29 proteins, and 50 fat J^−1^; 19.85 kJ gross energy g^−1^) for fourteen weeks to reproduce a model of overweight recently characterized for visceral adipose tissue inflammation [27].

Each group was divided into 2 subgroups (n = 5): five rats were subjected to food deprivation for 5 h and were subsequently injected with insulin (10 U/kg bodyweight; 5 rats) for the determination of insulin sensitivity by insulin tolerance test (ITT). The remaining five rats were used for all other measurements. The minimum sample size (n = 5) was calculated based on a G* Power test, which was performed using software obtained from the University of Dusseldorf (http://www.gpower.hhu.de/, accessed on 19 December 2021). The power was 0.90, the effect size (f) was 1.2249, and the α was 0.05.

During daily care, upon visual observation, all animals did not show significant behavioral changes. In particular, no pathological changes in water and food intake were observed, so that no animals ever showed weight loss or changed interaction with the accommodation environment. No specific behavioral tests were performed.

At the end of the treatment, the rats were anesthetized using an intraperitoneal injection of chloral hydrate (40 mg 100 g^−1^ BW) and blood was drawn via the inferior cava vein. Livers were excised and weighed. Liver samples were processed immediately for the preparation of mitochondria and other samples were frozen in liquid nitrogen and stored at −80 °C for further processing. Serum levels of cholesterol and triglycerides were determined using standard procedures. It is important to underline that, in rodents, sex-dependent differences in obesity susceptibility, hepatic lipid accumulation, and insulin sensitivity in response to diet exist. Specifically, male rats, although characterized by lower increases in body fat, typically show higher liver fat accumulation [28,29].

### 2.2. Indirect Calorimetry

Oxygen consumption (VO2) and carbon dioxide production (CO2) measurements were made using a four-chamber, indirect open-circuit calorimeter (Columbus Instrument), with one rat per chamber at a room temperature of 28 °C to evaluate basal metabolic parameters, as previously described [27]. 

### 2.3. Insulin Tolerance Test

For the ITT, rats were fasted for 5 h and were injected intraperitoneally with insulin (homolog rapid acting, 10 units/kg body wt. in sterile saline; Novartis, Basel, Switzerland). Samples of blood were collected: the first before the insulin load to determine basal insulinemia and the others at various times afterward (as indicated in Figure 1), and glucose values were determined by means of a glucose monitor (BRIO, Ascensia, NY, USA), calibrated for use with rats. The time window for observation was set at 50 min, as at this time, the blood glucose values, both in N and HFD rats, fell far below the values at time 0, and the animals appeared stressed, so no further blood samples were collected.

### 2.4. Liver Hystology

Hematoxylin–eosin (H&E) staining of liver sections was performed using standard protocols [30]. Livers were fixed, dehydrated, embedded in paraffin, sectioned, and stained with hematoxylin–eosin. Slides were examined by a Nikon Eclipse E80i light microscope (Nikon, Japan).

### 2.5. Immunohistochemical Staining for ADRP

Paraffin-embedded liver sections (4 mm) were incubated with anti-ADRP polyclonal antibody (clone GP40-mN1, Fitzgerald Industries International, Concord, MA, USA). The immunoreaction was detected with the avidin–biotin–peroxidase complex (ABC) method [31] using ABC Vectastain-Elite Kit (Vectastain ABC Elite Kit, Vector Labs, Burlingame, CA, USA), as described previously [30].

### 2.6. Mitochondrial H_2_O_2_ Release

Mitochondrial H_2_O_2_ production was measured by the Amplex Red/horseradish peroxidase method [32]. Liver mitochondria were isolated as previously described [33]. Horseradish peroxidase (2 U/mL) catalyzes the H_2_O_2_-dependent oxidation of non-fluorescent Amplex Red (80 μM) to fluorescent resorufin red [34]. Fluorescence was followed at an excitation wavelength of 540 ± 20 nm and an emission wavelength of 590 nm ± 20 using 96-well black plates and a fluorescence microplate reader (Tecan Infinite 200, Tecan, Männedorf, Switzerland). 

### 2.7. Genomic DNA Isolation 

Total liver DNA was extracted using the Genomic-tip 20/G kit (Qiagen, Valencia, CA, USA) according to the manufacturer’s protocol. The quantification of the purified genomic DNA and PCR products was determined on Nanodrop One (Thermo Fisher Scientific, Waltham, MA, USA). 

### 2.8. Quantitative Polymerase Chain Reaction (QPCR) 

QPCR was performed on liver DNA extracts, as previously described [35,36], with the following modification: the PCR amplification was performed using the Ranger DNA Polymerase with the appropriate premixes (Bioline Ltd., London, UK). Two pairs of PCR primers were employed: 

mtDNA long fragment (13.4 Kbp): 5′-AAAATCCCCGCAAACAATGACCACCC-3′ (sense)/5′-GGCAATTAAGAGTGGGATGGAGCCAA-3′ (anti-sense);

mtDNA short fragment (235 bp): 5′-CCTCCCATTCATTATCGCCGCCCTGC-3′ (sense)/5′-GTCTGGGTCTCCTAGTAGGTCTGGGAA-3′ (anti-sense). 

QPCR products were quantified using PicoGreen dye (Invitrogen, Milan, Italy) and a fluorescence plate reader in the same manner as the template DNA. The resulting values were converted to relative lesion frequencies per 10 Kbp DNA by applying the Poisson distribution. 

### 2.9. Quantification of mtDNA Copy Number by Real-Time PCR 

Relative mtDNA copy numbers were measured by real-time quantitative PCR (qRT-PCR) using QuantStudio 5 System (Thermo Fisher Scientific, Waltham, Massachusetts, USA). The amplification of mitochondrial cytochrome c oxidase subunit II (COII, mitochondrial-encoded gene) and β-actin (nuclear-encoded gene) were examined. The primer sequences used were as follows: 

COII: 5′-TGAGCCATCCCTTCACTAGG-3′(sense)/5′-TGAGCCGCAAATTTCAGAG-3′(anti-sense); 

β-actin: 5′-CTGCTCTTTCCCAGATGAGG-3′(sense)/5′-CCACAGCACTGTAGGGGTTT-3′(anti-sense). 

In our study, the average Ct values of nuclear DNA and mtDNA were obtained for each case. The mtDNA content was calculated using ∆Ct = average Ct nuclear DNA—average Ct mtDNA and then by applying the formula mtDNA content = 2^(2∆Ct). 

### 2.10. Total RNA Isolation from Liver and qRT-PCR

Total RNA liver was isolated using TRIzol reagent (Invitrogen) according to the manufacturer’s protocol as previously described [27]. A quantity of 1 µg of RNA was used to synthesize cDNA strands in 20 µL reaction volume, according to the Quanti Tect Reverse Transcription Kit instructions (Qiagen, Hilden, Germany). 

The qRT-PCR was carried out with 50 nM gene-specific primers, IQ SYBR Green supermix (Bio-Rad, Hercules, CA, USA), and cDNA samples (2 µL) in a total volume of 25 µL. A melting curve analysis was completed following amplification from 55 to 95 °C to assure product identification and homogeneity. Tissue expression levels were repeated in triplicate and were normalized to the housekeeping gene β-actin (stable under our experimental conditions, PCR primers were designed by using the Primer 3 program [37], synthesized, and verified by sequencing at Eurofins Genomics (Ebersberg, Germany). Primers used were as follows: 

Polγ S: 5′-CTCCTACCTGCCTGTCAACC-3′, AS: 5′-GCTCCATCAGCGACTTCTTC-3′; 

Nrf1 S: 5′-TTGATGGACACTTGGGTAGC-3′, AS: 5′-GCCAGAAGGACTGAAAGCAG-3′; 

Top1mt S: 5′-CCAAGGTGTTTCGGACCTAC-3′, AS:5′-GTTTGCCCGGTTGTAAGCTA-3′; 

Ssbp1 S: 5′-AGCCAGCAGTTTGGTTCTTG-3′, AS: 5′-ATCGCCACATCTCATTTGTT-3′; 

Twinkle: 5′-AAGGAAGTGGCGGAGAGAC-3′, AS:5′-TGGTAAGGCCAAACATCACA-3′;

Ogg1 S: 5′-GACTCAGACCGAGGATCAGCTC-3′, AS: 5′-GCTATAGAGCTGAGTCAGGCTGAC-3′;

Tfam S: 5′-CAGAGTTGTCATTGGGATTGG-3′, AS: TTCAGTGGGCAGAAGTCCAT-3′;

Pgc1α S: 5′-GGAGCAATAAAGCAAAGAGCA-3′; AS: GTGTGAGGAGGGTCATCGTT-3′; 

Apex S: CAGATCAGAAAACGTCAGCCAG-3′; AS: GGTCTCTTGGAGGCACAAGATG-3′; 

CI subunit NDUFB8 S: 5′-ATGGTGACTACCCGATGCTC-3′, AS: 5′-CTGGGTGGTCCCATTCATAC-3′;

CII-30kDa S: 5′-GCAGGACCCCTTCTCTCTCT-3′, AS: 5′-TTCCTGGATTCAGACCCTTG-3′; 

CIII-Core protein 2 S: 5′-GCTTTGGTTGGACTTGGTGT-3′, AS: 5′-GCCACCCCTAATGTTGAGAA-3′; 

CIV subunit I S: 5′-CGGCCGTAAGTGAGATGAAT-3′, AS: 5′-GCAGGGATACCTCGTCGTTA-3′;

CV alpha subunit S: 5′-TTCAATGATGGGACTGACGA-3′, AS: 5′-TCTTCACCAACTGAGCAACG-3′;

VEGFR2: S: 5′-AAGCAAATGCTCAGCAGGAT-3′, AS: 5′- TAGGCAGGGAGAGTCCAGAA-3′;

B-actin S: 5′-TGTGTTGTCCCTGTATGCCT-3′, AS: 5′-CCCTCATAGATGGGCACAGT-3′.

### 2.11. Western Blot Analysis

Samples of rat liver were homogenized as previously described [38]. Liver lysates containing 30 µg protein were loaded in each lane and were electrophoresed on SDS-PAGE gels and transferred to nitrocellulose membrane. Membranes were blocked and after probed with the following antibodies: anti-PGC1α (Millipore, Burlington, MA, USA), anti-NRF1 (Abcam, Cambridge, UK), anti-TFAM (Abcam), anti-TOM20 (Cell signaling), anti-GPX4 (Abcam), anti-PRDX3 (Abcam), anti-SOD2/MnSOD (Abcam), anti-CAT (Abcam), anti-DRP1 (Abcam), anti-OPA1 (Abcam), anti-MNF2 (Abcam), anti-PAMPK Thr^172^ (Cell signaling, Danvers, MA, USA), anti-AMPK (Cell signaling), anti-PULK Ser^555^ (Cell signaling), anti-ULK1 (Cell signaling), anti-AMBRA1 (Cell signaling), anti-PINK1 (Cell signaling ) anti-PARKIN (Cell signaling), anti-LC3B (Novus biologicals), anti-POLγ (Novus biologicals, Bio-Techne SRL, Milano, Italy), anti-Total OXPHOS complexes cocktail (Abcam), and anti-βACTIN (Sigma Aldrich, St. Louis, MO, USA). As secondary antibodies, peroxidase anti-rabbit IgG (Vector Laboratories Burlingame, CA, USA) and peroxidase anti-mouse IgG (Vector Laboratories) were used. Horseradish peroxidase-conjugated secondary antibodies were detected with enhanced chemiluminescence.

### 2.12. Statistical Analysis 

Results are expressed as means ± SEM. Statistical analyses were performed using a two-tailed, unpaired Student’s *t*-test. Differences were considered statistically significant at *p* < 0.05. 

## 3. Results

### 3.1. Establishment of the MAFDL Rat Model 

Eight-weeks-old male rats, housed at thermoneutrality, were fed an HFD for 14 weeks. At the end of this period, they were characterized for metabolic phenotype compared to the standard-diet-fed controls. The HFD-fed rats gained about 67% more weight and showed a significantly greater energy intake (+33%) than N rats (Table 1). When compared to the tissue weight of N controls, in the HFD-fed rats, liver weight was increased by about 28%, white adipose tissue (WAT) weight was increased by about 183%, and the ratio of WAT weight (g)/ bodyweight (g) was increased by about 146% (Table 1). No significant differences were observed between the groups in the ratio of liver weight/ bodyweight. HFD-fed rats also showed higher serum levels of cholesterol and triglycerides (+92% and +126% vs. N, respectively), and a significant decrease in the respiratory quotient, indicating an altered serum lipid profile on one hand and a shift of metabolism toward the utilization of lipids, on the other (Table 1). 

The insulin tolerance test revealed impaired insulin responsiveness in HFD rats (Figure 1).

When the liver sections were stained by H&E, a clear lobular structure was shown; the hepatocytes, with round and central nuclei, appeared arranged in cords and were seen radiating from the central venules (V) to the portal areas (Figure 2a). In the liver parenchyma of N rats, only some hepatocytes showed a few small lipid droplets in the cytoplasm, whereas the liver of HFD rats showed an increased hepatic lipid accumulation, detected as numerous small but also many large lipid droplets.

To better investigate alterations of hepatic lipid droplets (LDs) accumulation in HFD experimental conditions, ADRP, a classical steatosis marker, was immunolocalized (Figure 2b). In the liver of N rats, little or no ADRP staining was present in the hepatocytes. By contrast, the liver of HFD rats exhibited numerous and large ADRP-positive vesicles, preferentially located in the cytoplasm of poor oxygenate hepatocytes, around the central vein. All the above results indicate that our rat model, based on the combination of both HFD feeding and housing at thermoneutrality, has developed all the key features of MAFLD. 

### 3.2. Mitochondrial ROS Production and Antioxidant Enzymes

To study the cellular and subcellular functional correlates of the so-far described metabolic features, markers of the redox balance were evaluated at the liver level. Mitochondrial H_2_O_2_ release, an indirect index of mitochondrial superoxide production in vitro, was determined in liver samples of N and HFD rats. When compared to N controls, HFD animals showed a significantly higher mitochondrial H_2_O_2_ release (1.91-fold change vs. N) (Figure 3a).

Antioxidant enzymes such as superoxide dismutases 2 (SOD2), glutathione peroxidase 4 (GPX4), peroxiredoxin 3 (PRDX3), and catalase (CAT) play a key role in protecting cells from oxidative damage by catalyzing the dismutation of superoxide anion to hydrogen peroxide, preventing membrane lipid peroxidation and detoxifying peroxides, and neutralizing hydrogen peroxide to non-radical ROS. As reported in Figure 3b, Western blot analysis revealed that the protein levels of SOD2 were significantly decreased (0.77-fold change) in the liver of HFD rats when compared to N group; in contrast, those of the GPX4 significantly increased (1.86-fold change), while those of PRDX3 and CAT remained unchanged (Figure 3b). The observed differences between N and HFD rats, as far as it concerns mitochondrial ROS production and antioxidant enzyme protein levels, suggest oxidative damage in the liver of the HFD animals.

### 3.3. mtDNA Copy Number and Expression of Markers involved in Mitochondrial Biogenesis and Replisome

Oxidative stress and mitochondrial function/dysfunction are intrinsically linked to each other. To investigate mitochondrial features in the rat model of MAFLD under study, key players of mitochondrial quality control mechanisms were evaluated. The mtDNA copy number, a marker of mitochondrial biogenesis, was determined. As reported in Figure 4a, the HFD rats showed a reduced liver mtDNA copy number, (0.43-fold change vs. N), suggesting an impaired mitochondrial biogenesis in such group. To corroborate this hypothesis, putative changes in the expression levels of peroxisome proliferative-activated receptor gamma coactivator 1α (PGC-1α), nuclear respiratory factor 1 (NRF1), and mitochondrial transcription factor A (TFAM), all markers of mitochondrial biogenesis and of TOM20 (a marker of tissue mitochondrial content), were evaluated. In HFD rats, Pgc*1**α* and Tfam mRNA expressions were significantly reduced by 0.53- and 0.54-fold, respectively, compared to N. Nrf1 transcription was reduced by 0.77-fold (vs. N) but did not reach the statistical significance (Figure 4b). Western blot analysis revealed that protein levels of PGC1α, NRF1, TFAM, and TOM20 were all significantly reduced vs. N by 0.74-, 0.50-, 0.22-, and 0.49-fold, respectively (Figure 4c). 

Since defective mitochondrial biogenesis and mtDNA may lead to organ dysfunction due to insufficient mtDNA-encoded protein synthesis and inadequate energy transduction, we further investigated on the effects of HFD on the liver expression of key factors involved in mtDNA maintenance and synthesis, i.e., the minimum mitochondrial replisome composed of three nuclear genome-encoded proteins: the replicative mtDNA helicase Twinkle, DNA polymerase γ (Polγ), and the mitochondrial single-stranded DNA-binding protein (mtSsbp1). Twinkle, mtSsbp1, and Polγ mRNA levels were significantly lower in the HFD group (0.69-, 0.66-, and 0.42-fold change vs. N, respectively) (Figure 5a). HFD rats also showed significantly reduced POLγ protein representation levels (0.60-fold change vs. N, Figure 5b). The regulation of mtDNA, as well as replication, transcription, and translation, requires additional molecular factors, among which is the mitochondrial topoisomerase Top1mt. A qRT-PCR analysis revealed in HFD rats reduced mRNA expression levels of Top1mt (0.65-fold change vs. N), which, however, did not reach the statistical significance (Figure 5).

To further characterize the model under study, we evaluated the mRNA and protein levels of representative subunits of respiratory chain complexes CI–CV (Figure 5). In HFD rats, significant downregulations of CII, CIII, and CV transcriptional levels were observed (0.61-, 0.65-, and 0.61-fold change vs. N, respectively), while the mRNA expression of CI and CIV did not significantly change between the groups. Western blot analysis revealed a significant downregulation of CI, CIII, and CV protein levels (0.51-, 0.60-, and 0.58-fold change vs. N respectively), while CII and CIV levels were unchanged between the experimental groups.

### 3.4. Expression of Proteins Involved in Mitochondrial Dynamics

Mitochondria are highly organized and dynamic organelles that undergo continuous fission and fusion. The process of mitochondrial fusion is regulated by GTPase proteins, mitofusin 1 and 2 (MFN1/2), and optic atrophy 1 (OPA1), which are located in the outer and inner mitochondrial membranes, respectively. Furthermore, mitochondrial fission is controlled by dynamin-related protein 1 (DRP1). Overall, mitochondrial dynamics affect mitochondrial functions, ROS production, and mtDNA maintenance, thereby regulating the cellular fluxes of intramitochondrial contents. Thus, we next explored the effects of HFD in our MAFLD rat model, namely on the expression of markers related to mitochondrial dynamics. As reported in Figure 6, the protein levels of MFN2, but not those of DRP1 and OPA1, were significantly reduced (0.22-fold change) in the liver of the HFD rats when compared to the N animals. Even in the absence of significant changes to the protein representation levels of DRP1 and OPA1, these results suggest an imbalanced mitochondrial dynamic in MAFLD, likely in favor of fission.

### 3.5. Expression of Markers of Mitophagy 

Recent studies have revealed that one ancestral function of AMPK is to promote mitochondrial health [39], and multiple newly discovered targets of AMPK are involved in various aspects of mitochondrial homeostasis, including mitophagy, a crucial step in the cellular quality-control mechanisms, i.e., removing/recycling damaged or toxic organelles. AMPK may influence autophagy through the phosphorylation and direct activation of one of its earliest mediators, unc-51-like kinase 1 (ULK 1).

Our analysis showed that, in HFD rats, the phosphorylation/activation of AMPK on Thr172 and of ULK on Ser555 were both significantly increased by a 1.48-fold change when compared to N (Figure 7a,c). Upon activation, ULK1 phosphorylates several proteins involved in the execution of autophagy, including its binding partners and downstream effectors, among which is AMBRA1. A significantly increased expression of AMBRA1 (1.44-fold change vs. N) was observed in the liver homogenates of the HFD rats (Figure 7a,c), confirming the activation of autophagy in the group.

We next examined the representation levels of key proteins involved in the mitophagy flux: microtubule-associated protein 1A/1B-light chain 3B (LC3BII), PTEN-induced kinase 1 (PINK1), and the E3 ligase (PARKIN). In the liver of the HFD rats, a significant decrease in the expression of PINK1, PARKIN, and LC3BII (0.20-, 0.64- and 0.53-fold change vs. N, respectively) was observed (Figure 7b,c). These results indicate an impaired autophagic flux in the liver of the HFD rats.

### 3.6. mtDNA Damage and Lesion Frequency

Finally, we tested if impaired mitochondrial biogenesis, dynamics, and mitophagy contribute to an increased susceptibility to an accumulation of mtDNA damage. QPCR was used to measure the levels of hepatic mtDNA oxidative damage. In rats fed an HFD, the relative amplification of the long (13.4 Kbp) mtDNA fragment was reduced by 0.92-fold compared to N control rats (Figure 8a). Liver mtDNA from HFD rats contained significantly more mtDNA lesions compared to control rats (−0.31 vs. 0.028 lesion·10 Kbp^−1^) (Figure 8b). These results demonstrate that, in rats fed an HFD for 14 weeks, there was a significant increase in the oxidative damage of mtDNA and a significant reduction of its quality.

mtDNA repair mechanisms, in particular the base excision repair (BER) pathway, are important to maintain the integrity of the mtDNA itself. In addition to POLγ, it requires 8-oxoguanine glycosylase 1 (Ogg1) and apurinic/apyrimidinic endonuclease 1 (Ape1). A transcriptional analysis revealed significantly lower mRNA levels of Ogg1 and Ape1 in the liver of HFD rats (0.65- and 0.31-fold change vs. N, respectively; Figure 9). This reduction, together with the down-regulated expression observed for POLγ (Figure 5), suggests an impaired BER pathway in HFD rats.

## 4. Discussion

Obesity is considered the most important risk factor for hepatic steatosis and MAFLD, given that the worldwide increase in MAFLD prevalence and incidence has been linked with the rising trend in obesity [3]. Obesity is associated with the dysregulation of lipid homeostasis, which often leads to ectopic lipid deposition in non-adipocyte cells, such as skeletal muscle fibers and hepatocytes, and consequently to the development of insulin resistance [40]. Accumulating hepatocellular lipids are thought to simultaneously stimulate mitochondrial fatty acid oxidation and the production of ROS, thereby promoting lipid peroxidation and the damage of mtDNA and proteins [41]. Damaged mitochondria became dysfunctional, causing, in turn, a disruption in lipid homeostasis. The analysis of the so-far produced scientific literature reveals that the definition of the molecular mechanism underlying the alterations of the mitochondrial compartment in MAFLD/NAFLD has become as an important target of research as the mitochondrial quality control and its related operational mechanisms.

Our experimental set up produced overweight in male rats, an accumulation of lipids in the liver, a reduction of insulin responsiveness, and elevated plasma levels of cholesterol and triglycerides (Table 1; Figure 1 and Figure 2) thus confirming the MAFLD-state of the animals. 

Recent studies have indicated that a reduction of the hepatic mitochondrial function and abnormalities of lipid metabolism are associated with increases in ectopic fat deposition and insulin resistance in humans with MAFLD [42,43,44,45]. In particular, mitochondrial dysfunction includes impaired oxidative phosphorylation (OXPHOS), and increased production of ROS [46]. A gene expression analysis suggested that the observed mitochondrial dysfunction might be attributable to the downregulation of genes related to the mitochondrial quality control, which has been demonstrated to operate through the coordination of biogenesis, dynamics, and mitophagy to ensure cell homeostasis [47].

Impaired mitochondrial quality control results in the accumulation of damaged mitochondria that may release more ROS [48]. In a previous study, we demonstrated that rats receiving an HFD for 6 weeks exhibited an alteration in mitochondrial oxidative stress parameters, such as an increase in H_2_O_2_ production and an inhibition of aconitase and superoxide dismutase activity [49,50]. In a more recent study, we have demonstrated that rats fed an HFD for 14 weeks showed markedly increased serum levels of 8-OHdG, indicating a condition of increased systemic oxidative stress [27,51,52]. In line with the previous literature, data obtained in the present study show an increase in mitochondrial ROS production and an imbalance of antioxidant defense enzymes in the set-up model of the MAFLD (Figure 3). Under normal conditions, H_2_O_2_ is typically decomposed by GPX and catalase [53], while at high H_2_O_2_ concentrations, catalase is expected to greater contribute to its neutralization [54]. In our models of MAFLD, H_2_O_2_ accumulation was paralleled by increased protein levels of GPX alone, with an unchanged protein representation of CAT, this last condition likely exacerbating ROS production and stress. Moreover, we reported decreased protein levels of SOD2, which have been shown to be strictly correlated to increased DNA damage and increased incidences of cancer [55]. Vascular endothelial growth factor receptor 2 (VEGFR2) is one of the relevant so-far characterized SOD2-regulated genes and a primary responder to the vascular endothelial growth factor signal, regulating endothelial migration and proliferation. It is considered a potential cell type marker, and data on VEGFR2 expression have also been interpreted as having therapeutic significance [56]. The transcriptional analysis of VEGFR2 mRNA expression levels in our model revealed no significant changes between the groups. 

Of great relevance, is that in such conditions, mitochondrial components such as enzymes, carriers, respiratory chain components, and in particular the mitochondrial genome, may be affected. Alterations of the mitochondrial genome can be more specifically divided into two types: (1) decreased amount of mtDNA related to reduced mitochondrial mass and biogenesis, (2) mutations of mtDNA. These last occur at a much higher rate of those of the nuclear genome (about 100-fold higher) and important factors responsible for this feature have been recognized in the location of mtDNA in close proximity to OXPHOS, the main source of cellular ROS, its histone deficiency, and limited repair capabilities [57]. Based on our data (see Figure 3), it is evident that the oxidative stress induced by the long-lasting HFD regimen is associated in liver to mtDNA damage, in terms of both impaired mitochondrial biogenesis/turnover (decreased mtDNA copy numbers and TOM20 protein expression, decreased PGC1α, NRF1, and TFAM expressions), and an increased frequency of lesions and altered BER pathways (Figure 4, Figure 8 and Figure 9).

Among the mitochondrial biogenesis markers, TFAM plays a dual role: it maintains the mtDNA copy number by regulating mtDNA replication and it wraps mtDNA entirely to form a nucleoid structure, whose compactness depends on the local concentration of TFAM itself [58].

In the relaxed state, mtDNA is accessible for replication by the minimum mitochondrial replisome, which is formed by the hexameric DNA helicase TWINKLE, the tetrameric mtSSB, and the mtDNA polymerase POLγ [59]. Our results demonstrated that in the liver of MAFLD rats the significantly reduced expression levels of TFAM correspond to a decreased amount of the mtDNA copy number and an impaired expression of the key enzymes of the mitochondrial replisome machinery, further supporting the role of TFAM in the control of mtDNA levels in response to altered metabolic states [60,61]. 

Our MAFLD model is characterized not only by a suppression of the genes involved in the maintenance of mtDNA but also by reduced expression levels of representative subunits of the respiratory chain complexes. These results, in line with the literature, lead us to assume both functional and energetic mitochondrial damages [62]. Indeed, the decreased expression of respiratory chain complexes (CI, CIII, and CV) and the consequent impairment of the ability to synthesize ATP may give rise to several vicious cycles involving ROS, the respiratory chain polypeptides themselves, and mtDNA, which together with putative-increased lipogenesis and decreased fatty acid β-oxidation lead to fat-engorged liver and insulin resistance. Although the levels of ATP and AMP were not directly measured, the observed activation of AMPK in HFD rats seems to suggest reduced ATP production in favor of AMP in the liver of such animals.

Recent studies have revealed that obesity-associated enhanced ROS production can alter the DNA damage response through the inhibition of the expression of genes involved in the DNA repair [63,64,65,66]. Our data showed in the liver of MAFLD rats an increase in mitochondrial DNA damage and a decrease in the DNA repair mechanism, demonstrated by the significant reduction of the expression of the key players of the mitochondrial BER system.

All these conditions, likely indicative of altered mitochondrial functions, can be reflected in an unbalanced mitochondrial remodeling and network. Indeed, mitochondrial structural remodeling through mitochondrial dynamics, including fusion and fission, is intrinsically linked to the regulation of mitochondrial function [67,68,69]. Fusion connects neighboring depolarized mitochondria and mixes their contents to maintain membrane potential, while fission segregates damaged mitochondria from healthy ones and subjects them to mitophagy or fusion, respectively. In our study, the liver of MAFLD rats showed a severe reduction (0.22-fold change vs. N rats) of MFN2 protein representation (Figure 6), suggesting an imbalance of mitochondrial dynamics, likely towards fission. These results appear in accordance with those showing dysregulated mitochondrial dynamics as associated with the pathophysiology of metabolic diseases, such as obesity, NAFLD, and type 2 diabetes [65,70,71]. 

Mitochondrial homeostasis is also maintained by the balance between mitochondrial biogenesis and degradation, which is achieved through autophagy, a breakdown process to remove and recycle unwanted or damaged cellular components in lysosomes [72]. When macroautophagy selectively degrades mitochondria, it is termed mitophagy. Increasing lines of evidence suggest that mitophagy plays an important role in degrading damaged or unnecessary mitochondria in a variety of liver-related metabolic diseases [69]. Together with mitochondrial fission and fusion, and mitochondrial biogenesis, mitophagy is one of the most important steps necessary to maintain the quality control of mitochondria. Recent studies have revealed that a central role in mitophagy is played by AMPK [73].

We demonstrated an impaired mitophagy in the liver of MAFLD rats. In fact, while the increase in the protein representation of P-AMPK, P-ULK1, and AMBRA1 would indicate an activation of the autophagic pathway, the reduction in the protein expression of PINK1 and PARKIN together with that of PGC1α would indicate the failure of the mitophagic flux (Figure 7), as well as of biogenesis, respectively. The defective regulation of hepatocyte mitophagy and biogenesis may promote downstream effects, such as increased oxidative stress and inflammation, which, in turn, can contribute to the progression of liver diseases. Our results are in accordance with several previous studies. Defective mitophagy has been reported in both in vivo and in vitro models of NAFLD (e.g., HFD mice, cultured cells treated with oleic or palmitic acid), in which it has been associated with a series of NAFLD-related phenotypes, including increased fat accumulation, elevated oxidative stress, and inflammation [39,74,75,76]. A recent study has shown that the hepatocyte-specific deletion of PARKIN exacerbates fatty liver disease and insulin resistance in HFD-fed mice [77]. Hepatic mitophagy flux, measured in the mitophagy reporter mouse, mt-Keima, was reduced in obese mice with fatty liver, demonstrating that the loss of hepatic mitophagy is associated with NAFLD [78]. The results, showing the involvement of PINK1 and PARKIN in the coordination of the mitophagic flux, strengthen the notion that mitophagy may play an important and a protective role against the pathogenesis of hepatic steatosis. 

## 5. Conclusions

In conclusion, all together, our findings demonstrate a dysregulation of mitochondrial quality control in HFD-induced MAFLD in rats housed at thermoneutrality, consisting of DNA damage and reduced mechanisms of DNA repair. The obtained information may be relevant when targeting mitochondria for the management of liver dysfunctions in conditions of overweight, obesity, and related comorbidities. However, a limiting aspect for the present study concerns the interesting topic of the comparative analysis of the effects at different housing temperatures (22 °C vs. 28 °C—thermoneutrality), even if some interesting data present in the literature would justify the previous assertions. Indeed, it is documented that exposure to thermoneutrality worsens, in rodents, the metabolic response to increased caloric intake, while mild cold exposure reduces susceptibility to overweight and ectopic fat accumulation [79].

## Figures and Tables

**Figure 1 genes-13-00315-f001:**
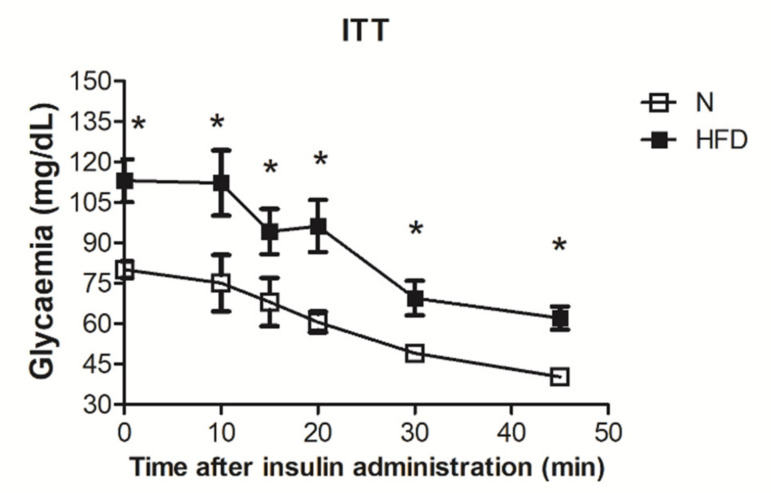
Insulin tolerance test in N and HFD rats: Glycaemia after insulin administration at different time points (0, 10, 15, 20, 30, and 45 min). Values are means ± SEM of five independent measurements (n = 5), * *p* < 0.05 vs. N; N: rats receiving a standard diet for 14 weeks; HFD: MAFLD rats receiving an HFD for 14 weeks.

**Figure 2 genes-13-00315-f002:**
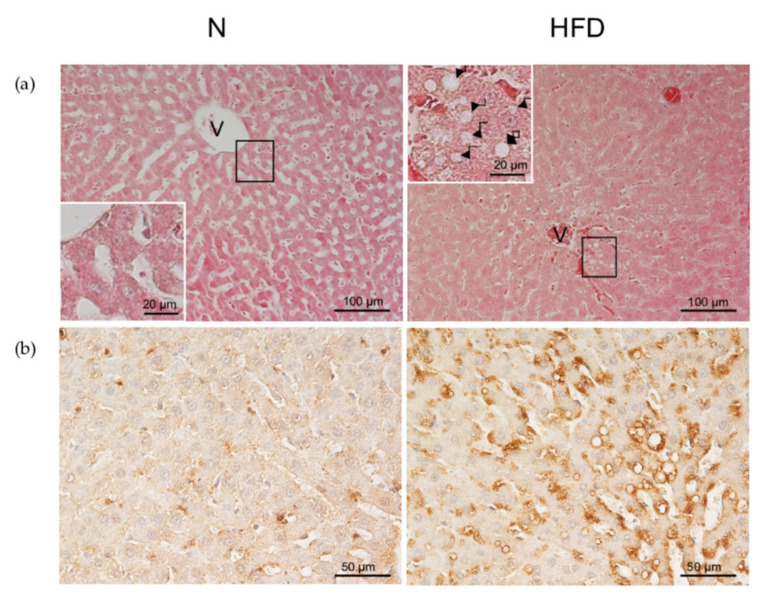
Hepatic tissue morphology in N and HFD rats: (**a**) H&E staining of the liver, paraffin- embedded sections. Numerous fatty droplets (indicated by arrows) were observed within hepatocytes in HFD rats. Enlargement of the squared areas of the panel revealed the presence of very large LDs (arrows) in the hepatocytes located around the central vein in the hepatic parenchyma of HFD rats (insets). (**b**) The presence of ADRP was detected by immunostaining. In the liver of N controls, a slight presence of ADRP-positive microvesicular lipid droplets was observed. In the liver of HFD rats, the ADRP staining was more intense and became even more marked in the hepatocytes around the central veins (V). N: control group; HFD: MAFLD rats receiving an HFD for 14 weeks.

**Figure 3 genes-13-00315-f003:**
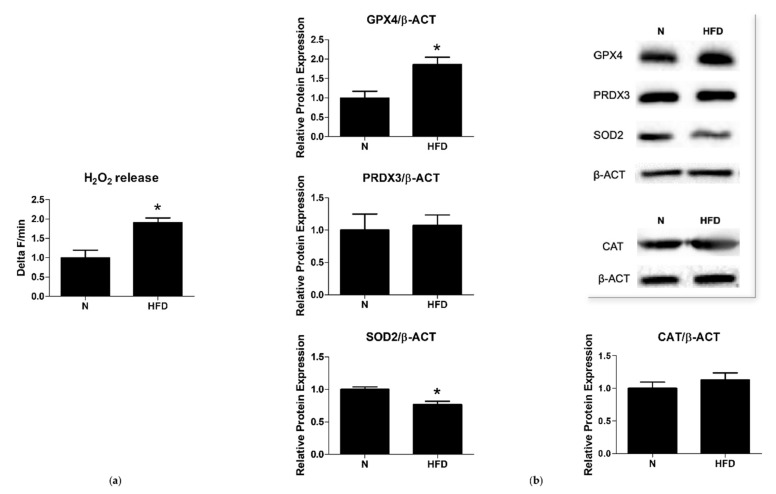
Mitochondrial H_2_O_2_ release and antioxidant enzymes: (**a**) Mitochondrial H_2_O_2_ release (Delta F/min) in liver of N and HFD rats. (**b**) Quantitative analysis and representative Western blot of GPX4, PRDX3, SOD2, and CAT protein levels in liver of N and HFD rats. Protein representation levels were normalized based on β-actin. Data were normalized to the value obtained for N animals, set as 1. Values are presented as means ± SEM (n = 5). N: control group; HFD: MAFLD rats receiving an HFD for 14 weeks. * *p* < 0.05 vs. N.

**Figure 4 genes-13-00315-f004:**
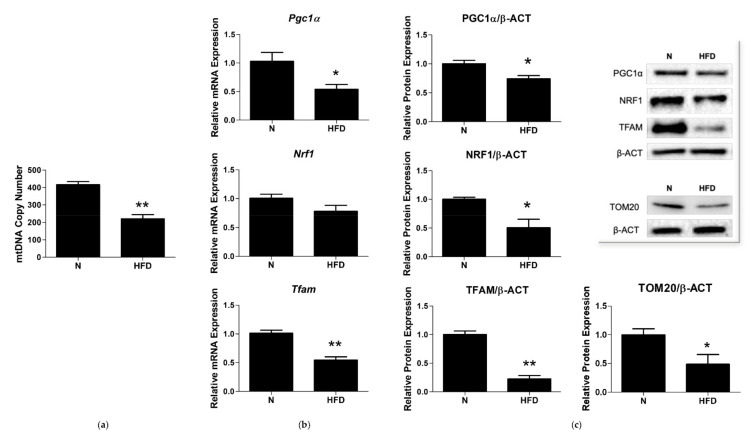
mtDNA copy number and mitochondrial biogenesis markers in liver of N and HFD rats: (**a**) mtDNA copy number was assessed by qRT-PCR in 10 ng of genomic liver DNA using primers for mtCOII and β-actin, as reported in the Materials and Methods section; (**b**) Pgc1α, Nrf1, and Tfam mRNA expressions were determined by qRT-PCR; (**c**) quantitative analysis and representative Western blot of PGC1α, NRF1, TFAM, and TOM20 protein levels. Protein representation levels were normalized based on β-actin. Data were normalized to the value obtained for N animals, set as 1. Values are presented as means ± SEM (n = 5). N: control group; HFD: MAFLD rats receiving an HFD for 14 weeks. * *p* < 0.05, ** *p* < 0.01 vs. N.

**Figure 5 genes-13-00315-f005:**
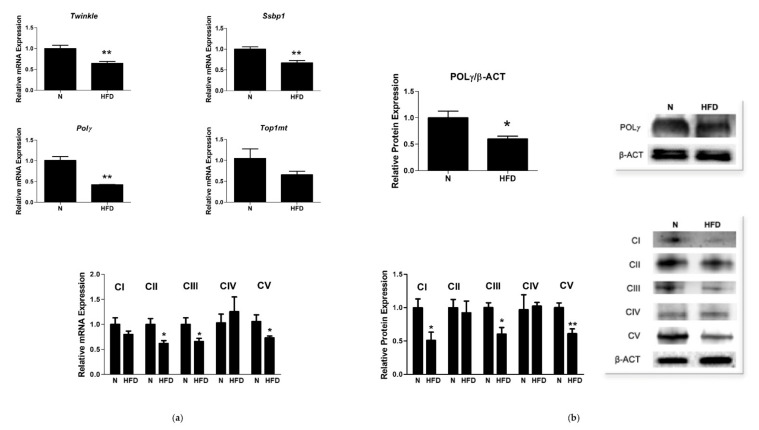
Mitochondrial replisome in liver of N and HFD rats: (**a**) Twinkle, Ssbp1, Polγ, TOP1mt, and CI–CV respiratory chain complex mRNA-expressions were determined by qRT-PCR. (**b**) Quantitative analysis and representative Western blot of POLγ and CI–CV respiratory chain complex protein levels. Protein representation levels were normalized based on β-actin. Data were normalized to the value obtained for N animals, set as 1. Values are presented as means ± SEM (n = 5). N: control group; HFD: MAFLD rats receiving an HFD for 14 weeks. * *p* < 0.05, ** *p* < 0.01 vs. N.

**Figure 6 genes-13-00315-f006:**
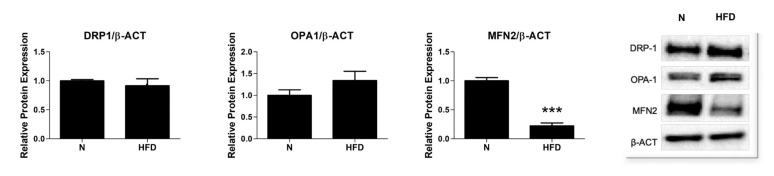
Mitochondrial dynamics in liver of N and HFD rats. Quantitative analysis and representative Western blot of DRP1, OPA1, and MNF2 protein levels. Protein representation levels were normalized based on β-actin. Data were normalized to the value obtained for N animals, set as 1. Values are presented as means ± SEM (n = 5). N: control group; HFD: MAFLD rats receiving an HFD for 14 weeks. *** *p* value 0.001 vs. N.

**Figure 7 genes-13-00315-f007:**
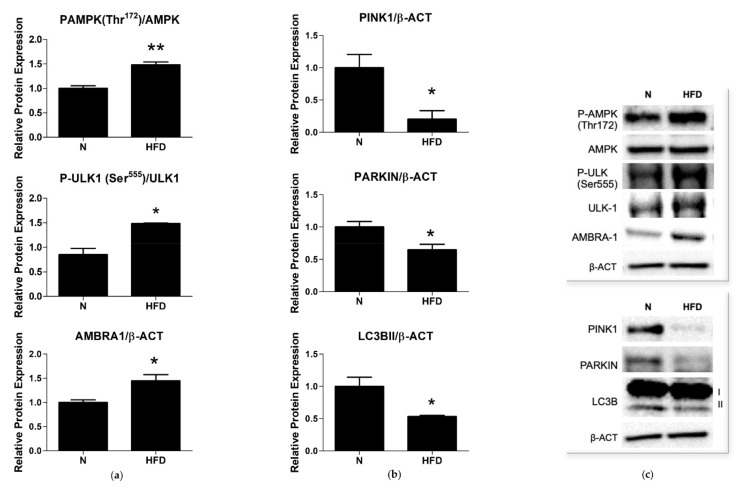
Selected markers of autophagy/mitophagy: (**a**,**c**) Quantitative analysis and representative Western blot of PAMPK, P-ULK1, and AMBRA1 protein levels; (**b**,**c**) quantitative analysis and representative Western blot of PINK1, PARKIN, and LC3B2 protein levels. Protein representation levels were normalized based on β-actin. Data were normalized to the value obtained for N animals, set as 1. Values are presented as means ± SEM (n = 5). N: control group; HFD: MAFLD rats receiving an HFD for 14 weeks. * *p* < 0.05, ** *p* < 0.01 vs. N.

**Figure 8 genes-13-00315-f008:**
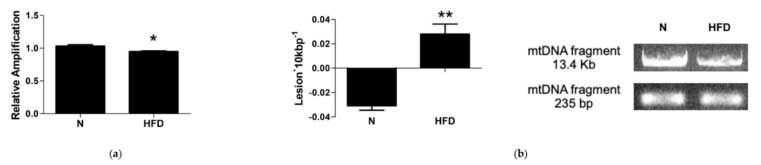
mtDNA damage and lesion frequency in liver of N and HFD rats: (**a**) mtDNA damage was evaluated by amplifying long (13.4 Kbp) and short (235 bp) mtDNA fragments by QPCR; (**b**) frequency of mtDNA lesions per 10 Kbp per strand. Values are presented as means ± SEM (n = 5). N: control group; HFD: MAFLD rats receiving an HFD for 14 weeks. * *p* < 0.05, ** *p* < 0.01 vs. N.

**Figure 9 genes-13-00315-f009:**
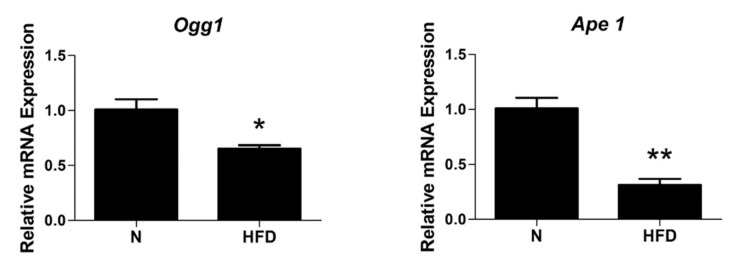
Base excision repair (BER) in liver of N and HFD rats: Ogg1 and Ape1 mRNA expressions were measured by qRT-PCR. Values are presented as means ± SEM (n = 5). N: control group; HFD: MAFLD rats receiving an HFD for 14 weeks. * *p* < 0.05, ** *p* < 0 .01 vs. N.

**Table 1 genes-13-00315-t001:** Bodyweight gain, liver weight, liver weight/ bodyweight ratio, white adipose tissue (WAT) weight, WAT weight/ bodyweight ratio, energy intake, respiratory quotient, in N and high-fat diet (HFD) rats.

Parameters	N	HFD
Bodyweight gain (g)	115 ± 12.65	192.30 ± 8.18 *
Liver weight (g)	7.48 ± 0.23	9.59 ± 0.36 *
Liver weight (g)/ Bodyweight (g)	0.021 ± 0.00029	0.022 ± 0.00032
WAT weight (g)	11.20 ± 0.98	31.70 ± 1.80 *
WAT weight (g)/ Bodyweight	0.030 ± 0.0015	0.074 ± 0.0023 *
Energy intake (KJ)	5098 ± 408	6798 ± 498 *
Respiratory quotient	0.99 ± 0.058	0.72 ± 0.013 *
Triglycerides (mg/dL)	115 ± 5.45	260.50 ± 17.33 *
Cholesterol (mg/dL)	39 ± 4.39	74.70 ± 3.11 *

Notes: Values are means± SEM of five independent measurements (n = 5), * *p* < 0.05 vs. N; N: rats receiving a standard diet for 14 weeks; HFD: MAFLD rats receiving an HFD for 14 weeks.

## Data Availability

The original contributions presented in the study are included in the article. Further inquiries can be directed to the corresponding authors.

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
