# Peer review of "Altered Mitochondrial Quality Control in Rats with Metabolic Dysfunction-Associated Fatty Liver Disease (MAFLD) Induced by High-Fat Feeding"

_genes, 2022, doi:10.3390/genes13020315_

Round 1
Reviewer 1 Report
To the authors,
The article “Altered mitochondrial quality control in rat with metabolic dysfunction-associated fatty liver disease (MAFLD) induced by 3 high-fat feeding” is a clear and well-structured manuscript. However, some of the reported results are not as clear as the author claims.
- Some of the intensity bands of the western-blot are similar, and the authors claim a decrease in the protein expression level. Some examples are figure 3b GPX4, PRDX3 and SOD2 proteins, figure 4c PGC1, NRF1 and TFAM (it does not seem to have a 77% decrease in protein level expression) proteins, figure 5b POLγ protein, and figure 7c LC3B protein.
Please, provide data that confirm your affirmations or remove these figures from the manuscript and rewrite the conclusion derived from these results.
- In the discussion section, some of the conclusions are made from the WB results, which are not as clear as the author claims, so please rewrite the section according to the real data.
For instance in lines 497 to 500 it is true that there is an increase in ROS production but the protein level of GPX4, PRDX3 and SOD2 do not really change, so you have not confirmed an imbalance of the antioxidant defenses. In order to confirm these I strongly recommen the measure of the activity level of these antioxidant enzymes involved in this process.
The same happens with others assertions made in lines 511-512 (decrease of PGC1-a NRF1 and a 77% decrease in TFAM is not as clear as the author says).
Please recheck everything carefully or provide data that really confirm the decrease in the protein level in the HFD group compared to control.
Furthermore, there are some minor errors, which should be corrected:
- In line 97 there is a missing bracket before “Muscedola”
- In line 103 there are two “J”, please remove one
- In line 122 change 1100 and 1600 to 11 and 16 or 11:00 and 16:00
- In line 132 you referenced a table which is not in the manuscript. Please remove the sentence or provide the table.
- QPCR should be change to qPCR in lines 167, 168,182, 440 and 451.
- Swap “Total liver RNA” to “Total RNA liver” in line 204
- Which means WAT in line 277?, please define it
- Notes from the table 1 should be immediately after the table and not after the figure 1
- Please define the meaning of “H&E” in line 294
- Write in italics “in vitro” in line 319
- The result in figure 2b reported changes in protein level of ADRP not from SOD2 (line 326). Please correct it
- In line 351 Pgc1α and Tfam should be in italics if they refers to gene names
- Write PGC1a or PGC-1α; please homogenize the writing. This includes the change of PGC1 to PGC1a in line 361
- You talk about “recent studies about the ancestral function of AMPK” but you do not cite any of them. Please provide the proper references (line 411)
- Correct/swap the data relate to mtDNA lesions from line 444
- In line 456 Ogg1 and Ape1 should be in italics if they refers to gene names
- In lines 454 and 458 you refers to POLG instead of POLγ. Moreover, in line 458 the figure you should made reference is figure 5 not 4.
- MFN2 is written as “Mfn2” in line 546 and the same happen with PARKIN (“Parkin”) in line 568, please homogenize the writing of them
- Figure 6 in line 563 it is figure 7
- Write in italics “in vivo” and “in vitro” in line 565
- In the legend of the figures 1 to 3 there is a single line spacing but in the legend of the figures 4 to 9 is a double line spacing. Please homogenize them as well as the space between the paragraphs and between some words.
Best regards,
Author Response
- Some of the intensity bands of the western-blot are similar, and the authors claim a decrease in the protein expression level. Some examples are figure 3b GPX4, PRDX3 and SOD2 proteins, figure 4c PGC1, NRF1 and TFAM (it does not seem to have a 77% decrease in protein level expression) proteins, figure 5b POLγ protein, and figure 7c LC3B protein. Please, provide data that confirm your affirmations or remove these figures from the manuscript and rewrite the conclusion derived from these results.
Reply: According to the Reviewer's suggestion, we have changed the representative western blot of SOD2, TFAM, POLγ and LC3B proteins (see, Fig. 3b, Fig. 4c, Fig. 5b and Fig. 7c of the revised MS) better representing the obtained quantitative data. In addition, to give further strength to the reduction of mitochondrial biogenesis observed in HFD rats (also following one of the comments of the Reviewer 3), we added data concerning protein expression levels of TOM20, being this protein a recognized marker of tissue mitochondrial content (see, Figure 4c of the revised MS).
- In the discussion section, some of the conclusions are made from the WB results, which are not as clear as the author claims, so please rewrite the section according to the real data. For instance in lines 497 to 500 it is true that there is an increase in ROS production but the protein level of GPX4, PRDX3 and SOD2 do not really change, so you have not confirmed an imbalance of the antioxidant defenses. In order to confirm these I strongly recommend the measure of the activity level of these antioxidant enzymes involved in this process. The same happens with others assertions made in lines 511-512 (decrease of PGC1-a NRF1 and a 77% decrease in TFAM is not as clear as the author says).
Reply: We thank the reviewer for her/his constructive criticism. To confirm the imbalance of the antioxidant defenses, we analyzed also catalase protein expression levels (see, Figure 3b of the revised MS). In the liver, catalase is known to have an excellent ability to remove H2O2, even compared to GPX. In our hands, although the stronger increase of H2O2 levels in the liver of HFD rats, catalase expression failed to significantly change, likely exacerbating/causing oxidative stress in MAFLD conditions. Consequently, the authors revised the parts of the text in the Results and Discussion sections dedicated to the altered expression of antioxidant enzymes (see, Results, lines 331-341 and Discussion, lines 511-518, of the revised MS). Unfortunately, in the time of the preparation of this response we were unable to measure antioxidant enzyme activities due to lack of appropriate reagents and available biological samples.
- Furthermore, there are some minor errors, which should be corrected:
In line 97 there is a missing bracket before "Muscedola"
In line 103 there are two "J", please remove one
In line 122 change 1100 and 1600 to 11 and 16 or 11:00 and 16:00
In line 132 you referenced a table which is not in the manuscript. Please remove the sentence or provide the table.
QPCR should be change to qPCR in lines 167, 168,182, 440 and 451.
Swap "Total liver RNA" to "Total RNA liver" in line 204
Which means WAT in line 277?, please define it
Notes from the table 1 should be immediately after the table and not after the figure 1
Please define the meaning of "H&E" in line 294
Write in italics "in vitro" in line 319
The result in figure 2b reported changes in protein level of ADRP not from SOD2 (line 326). Please correct it
In line 351 Pgc1α and Tfam should be in italics if they refers to gene names
Write PGC1a or PGC-1α; please homogenize the writing. This includes the change of PGC1 to PGC1a in line 361
You talk about "recent studies about the ancestral function of AMPK" but you do not cite any of them. Please provide the proper references (line 411)
Correct/swap the data relate to mtDNA lesions from line 444
In line 456 Ogg1 and Ape1 should be in italics if they refers to gene names
In lines 454 and 458 you refers to POLG instead of POLγ. Moreover, in line 458 the figure you should made reference is figure 5 not 4.
MFN2 is written as "Mfn2" in line 546 and the same happen with PARKIN ("Parkin") in line 568, please homogenize the writing of them
Figure 6 in line 563 it is figure 7
Write in italics "in vivo" and "in vitro" in line 565
In the legend of the figures 1 to 3 there is a single line spacing but in the legend of the figures 4 to 9 is a double line spacing. Please homogenize them as well as the space between the paragraphs and between some words.
Reply: According to the Reviewer's suggestions, all corrections have been made and highlighted in red (see, revised MS). As far as it concerns the abbreviation "QPCR" it was not changed in "qPCR". Indeed, it refers to a different method compared with “real time Pcr”, applied to DNA and not to RNA, as described in Materials and Methods section (see, Materials and methods 2.6 Quantitative Polymerase Chain Reaction (QPCR), lines 172-188 of the revised MS).
Reviewer 2 Report
The manuscript entitled “Altered mitochondrial quality control in rat with metabolic dys-function-associated fatty liver disease (MAFLD) induced by high-fat feeding” submitted by Federica Cioffi show how deposition and liver steatosis impact mitochondrial quality control (ROS, mtDNA, mitochondrial biogenesis, etc.). The topic has high interest for the scientific community, and the article is well-written. All the methods are explained in detail, and the figure legends are complete.
- Only H202 levels are measured, and SOD2 levels are lower in HFD rats. Since SOD2 is responsible for H2O2 production, further exploring why H2O2 levels are higher is necessary. I hardly recommend looking at other antioxidant enzymes like catalase, responsible for H2O2 elimination. Measurement of superoxide levels in the mitochondrial would also be relevant.
- Although the results are convincing, the mitochondrial biomass was never studied. I suggest monitoring mitochondrial biomass by fluorescence using the Mitotracker DeepRed reagent (or similar), or at least using a marker of mitochondrial content in your western blots (TOM20)
- Figure 5: I miss the protein levels of most of the proteins. For example, it is possible to look at the 5 complexes of the electron transport chain using a commercial cocktail antibody.
- It should be discussed the limitation of using only male rats and not females
Author Response
- Only H202 levels are measured, and SOD2 levels are lower in HFD rats. Since SOD2 is responsible for H2O2 production, further exploring why H2O2 levels are higher is necessary. I hardly recommend looking at other antioxidant enzymes like catalase, responsible for H2O2 elimination. Measurement of superoxide levels in the mitochondrial would also be relevant.
Reply: According to the reviewer's comment, protein levels of catalase were analyzed and results were included in the revised version of the manuscript (see, Figure 3b, Results, lines 331-341). Appropriate bibliographic references were also cited in the Discussion (see, Discussion section, lines 511-518 of the revised MS). Unfortunately, in the time of the preparation of this response we were unable to measure superoxide levels in the mitochondrial fraction due to lack of appropriate reagents and available biological samples.
- Although the results are convincing, the mitochondrial biomass was never studied. I suggest monitoring mitochondrial biomass by fluorescence using the Mitotracker DeepRed reagent (or similar), or at least using a marker of mitochondrial content in your western blots (TOM20)
Reply: As suggested by the reviewer, protein representation levels of TOM20 were also analyzed. The obtained results were included in the revised version of the manuscript (see, Figure 4c, Results, lines 357-366 and Discussion, lines 535-539 of the revised MS).
- Figure 5: I miss the protein levels of most of the proteins. For example, it is possible to look at the 5 complexes of the electron transport chain using a commercial cocktail antibody.
Reply: As suggested by the reviewer, by using a commercial cocktail of antibodies (ab110413, Total OXPHOS Rodent WB Antibody Cocktail, abcam) the relative protein levels of OXPHOS complexes (Complex I-V subunits) were analyzed by western blot. Results were added and discussed (see, Figure 5b, Results section, lines 392-394 Discussion lines 554-560 of the revised MS).
- It should be discussed the limitation of using only male rats and not females
Reply: According to the reviewer's suggestion, since only male rats were used, an explanation sentence has been added in the Materials and Methods section (see, Materials and Methods, lines 116-119 of the revised version). Indeed, sex-dependent differences in obesity susceptibility, hepatic lipid accumulation and insulin sensitivity in response to diet exist, with male rats typically being characterized by lower increase in body fat and higher liver fat accumulation. Accordingly, references have been up-dated (see, ref. 28 and 29 of the revised MS).
Reviewer 3 Report
This manuscript by Cioffi et al. has studied the effect of a high-fat diet on liver mitochondrial function in rats. Authors have investigated several key and quality control aspects of mitochondria and showed alteration caused by a high-fat diet after 14 weeks of continuous feeding. One of the major emphases by authors in this study was to maintain animals at thermoneutral zone. Some of my concerns are listed below.
- Authors have mentioned that using thermoneutral temperature should be used rather than standard temperature to better translate the results compared to humans. But the authors have not shown any comparison between standard temperature and thermoneutral temperature in terms of HFD and mitochondrial quality control. This is a significant limitation of this study.
- How did the authors choose a 14 week HFD window? Please provide a rationale.
- In fig 1, glycemia level was measured up to 50 minutes only. This time is a very short window; what happens beyond this time-point (like 120 min)?
- Figure 3, the decrease in SOD2 shown by western is not very convincing. What happens to SOD2 target genes like VEGFR2? A minor comment: the graph shows relative protein expression in fold change, but the text mentions it as percent change. Make it consistent, either fold change or percent change in all figures.
- Figure 7, HFD shows an increase in AMPK and ULK phosphorylation, which subsequently favors mitophagy. But HFD leads to a decrease in PGC1a (promote biogenesis), and at the same time, the level of PINK1 also decreases, which is inhibitory to biogenesis. So these changes need to be explained by looking at some more downstream effects.
- Does HFD affect mitochondrial activity in terms of ATP, AMP/ATP ratio?
- Do HFD and mitochondrial impairment show any behavioral changes?
Minor comments:
- Spelling error: line 136, hystology.
- Expand abbreviation WAT.
Author Response
- Authors have mentioned that using thermoneutral temperature should be used rather than standard temperature to better translate the results compared to humans. But the authors have not shown any comparison between standard temperature and thermoneutral temperature in terms of HFD and mitochondrial quality control. This is a significant limitation of this study.
Reply: We fully agree with the reviewer and are aware that not having made a comparison between animals at standard room temperature and animals at thermoneutrality represents a limitation of the study, even if the literature documents that exposure to thermoneutrality worsens, in rodents, the metabolic response to increased caloric intake while mild cold exposure reduces susceptibility to overweight and ectopic fat accumulation. To report this data we need to do a complete serie of new experiments with animals in two different conditions. This presumably would take about a year. This interesting aspect could be a topic for future investigations. However, following the reviewer's suggestion, a sentence of self-criticism was inserted in the concluding part of the discussion (limitations of the study) (see, Discussion, lines 609-615, of the revised MS).
- How did the authors choose a 14 week HFD window? Please provide a rationale.
Reply: According to the reviewer's suggestion, the rationale to choose a 14 week HFD window has been made explicit and an appropriate bibliographic reference has been added (see, Material and Methods section, lines 102-103 of the revised version of the MS.
- In figure 1, glycaemia level was measured up to 50 minutes only. This time is a very short window; what happens beyond this time-point (like 120 min)?
Reply: The reviewer is right, 50 minutes observation appears a short window, however at this time glycaemia values, both in N and HFD rats, fell far below the values ​​at time 0, and the animals appeared stressed, so that we did not take any further blood samples. At the same time, it is important to underline that 30-60 minutes are considered standard time intervals to monitor glucose concentration in rodents in ITT tests.
- Figure 3, the decrease in SOD2 shown by western is not very convincing. What happens to SOD2 target genes like VEGFR2? A minor comment: the graph shows relative protein expression in fold change, but the text mentions it as percent change. Make it consistent, either fold change or percent change in all figures.
Reply: According to the reviewer’s comment, in Fig. 3b of the revised MS we showed a more representative panel for SOD2 protein levels. VEGFR2 mRNA expression levels were also measured by qRT-PCR. Gene expression of this SOD2 target did not change between the groups. The Discussion section has been accordingly up-dated (see, lines 518-526 of the revised version). Relative expression has been reported in fold change (see, revised version of the MS).
- Figure 7, HFD shows an increase in AMPK and ULK phosphorylation, which subsequently favors mitophagy. But HFD leads to a decrease in PGC1a (promote biogenesis), and at the same time, the level of PINK1 also decreases, which is inhibitory to biogenesis. So these changes need to be explained by looking at some more downstream effects.
Reply: According to the Reviewer suggestion, we modified the text in the Discussion section furnishing a more detailed explanation for the likely functional downstream effects of the reported changes in biogenesis and mitophagy factors (see, Discussion, lines 587-593 of the revised MS) .
- Does HFD affect mitochondrial activity in terms of ATP, AMP/ATP ratio?
Reply: We did not directly measure ATP and AMP levels in the livers of the animals under study. However, the combination of data concerning respiratory complex expression (specifically of CV) and AMPK activation strongly suggest reduced ATP production in favor of AMP in HFD. Western blot analysis of OXPHOS has been added in the revised version of the MS and the concern made by the reviewer has been stressed in the revised version of the Discussion (see, lines 554-560). Unfortunately, in the time of the preparation of this response we were unable to measure tissue levels of ATP and AMP, due to lack of appropriate reagents and available biological samples.
- Do HFD and mitochondrial impairment show any behavioral changes?
Reply: During daily care, upon visual observation, the animals of both the experimental groups did not show significant behavioral changes. In particular, no pathological changes in water and food intake were observed, so that no animals ever showed weight loss or changed interaction with the accommodation environment. However, no specific behavioral tests were performed.
- Minor comments:
Spelling error: line 136, hystology.
Expand abbreviation WAT.
Reply: We thank the Reviewer for her/his comments. Suggested corrections have been made and highlighted in red throughout the MS.
Round 2
Reviewer 1 Report
-
Reviewer 2 Report
The authors addressed all my concerns, therefore I support the publication of the manuscript in genes
Reviewer 3 Report
Authors have tried to answer most of the concerns by providing explanation and modifications in texts (mostly discussion), while few concerns were answered by providing additional data.
Some of the major criticisms were not answered by experiments/data. and authors have mentioned the unavailability of reagents and biological samples. During the time of pandemic, I understand the current situation of lab requirements.
Please include the reason and rationale provided to answer the criticisms in manuscript to make it suitable for publication.